# Neural circuits underlying habituation of visually evoked escape behaviors in larval zebrafish

Haleh Fotowat[1,2], Florian Engert[2]*

[1]Wyss Institute for Biologically Inspired Engineering, Harvard University, Boston, United States; [2]Department of Molecular and Cellular Biology, Harvard University, Cambridge, United States

**Abstract** Larval zebrafish that are exposed repeatedly to dark looming stimuli will quickly habituate to these aversive signals and cease to respond with their stereotypical escape swims. A dark looming stimulus can be separated into two independent components: one that is characterized by an overall spatial expansion, where overall luminance is maintained at the same level, and a second, that represents an overall dimming within the whole visual field in the absence of any motion energy. Using specific stimulation patterns that isolate these independent components, we first extracted the behavioral algorithms that dictate how these separate information channels interact with each other and across the two eyes during the habituation process. Concurrent brain wide imaging experiments then permitted the construction of circuit models that suggest the existence of two separate neural pathways. The first is a looming channel which responds specifically to expanding edges presented to the contralateral eye and relays that information to the brain stem escape network to generate directed escapes. The second is a dimming-specific channel that could be either monocular or binocularly responsive, and that appears to specifically inhibit escape response when activated. We propose that this second channel is under strong contextual modulation and that it is primarily responsible for the incremental silencing of successive dark looming-evoked escapes.

*For correspondence:
florian@mcb.harvard.edu

Competing interest: The authors declare that no competing interests exist.

## Editor's evaluation

This study presents a valuable finding on how visual stimuli are processed during the habituation of visually evoked escape behaviors. The evidence supporting the claims of the authors are convincing, although future study to show the connectivity and function of the neural circuits would be important to get more insights into this process. The work will be of interest to neuroscientists working on habituation and learning.

## Introduction

Habituation or reduced responsiveness to repeated stimulation is the simplest form of learning and is ubiquitous across the animal kingdom (*Rankin et al., 2009*). Much is known about the neural mechanisms underlying habituation in the context of sensori-motor behaviors that are driven by abrupt sensory stimuli, through a small set of neurons. One famous example is the habituation of the gill withdrawal reflex to a brief touch stimulus through the local network of neurons in the abdominal ganglion in *Aplysia* (*Glanzman, 2009*; *Kandel, 2001*; *Rankin et al., 2009*), and another well-studied phenomenon is the habituation of escape responses to quick tap stimuli in larval zebrafish through the Mauthner cell circuitry in the hindbrain (*Marsden and Granato, 2015*; *Roberts et al., 2016*). Less is known about neural mechanisms and computational algorithms that underlie habituation of behaviors

that get executed at distinctly longer time scales, and where the sensory stimuli vary more slowly in time.

Objects approaching on a collision course, such as a looming predator, are good examples of such stimuli the processing of which often involves extracting behaviorally relevant kinematic variables such as angular size, speed, and time remaining to collision based on the spatio-temporal structure of the sensory input (*Fotowat and Gabbiani, 2011*). In invertebrates such as locusts this computation is carried out across layers of the optic lobes and especially through three restricted dendritic fields of a pair of identified neurons, the Lobula Giant Movement Detectors (LGMD) (*Gabbiani et al., 1999*; *Peron et al., 2009*; *Jones and Gabbiani, 2010*; *Gabbiani et al., 2001*). Distinct features of the time-varying firing in these neurons are in turn extracted by the downstream motor networks to drive the execution of a multi-stage escape behavior (*Fotowat et al., 2011*). In vertebrates such as fish, neurons tuned to looming stimuli are found in the optic tectum and are thought to perform computations similar to those carried out by the LGMD neurons, but these vertebrate animals tend to use a more distributed network of neurons (*Dunn et al., 2016*; *Temizer et al., 2015*). Information is then relayed from the visual system to the downstream motor networks which can trigger distinct types of escapes in a context-dependent manner (*Bhattacharyya et al., 2017*). Specifically, the two Mauthner cells, a pair of identified giant neurons in the hindbrain, further encode the course of stimulus approach in their time-varying subthreshold membrane potential, and will trigger a fast escape once a voltage threshold is crossed (*Preuss et al., 2006*). The time-varying activity in the distributed network of sensory and motor neurons thus encodes whether or not fast escapes should occur, in which direction, and when. Escaping too early or prematurely is undesirable as it will give enough time for a predator to revise its attack strategy, or it might not be necessary if the detected approaching object is in fact not an attacking predator. Escaping too late is also detrimental for obvious reasons.

In addition to being precise in timing and directionality, it is important for these behaviors to be flexible, as well as context and experience dependent. For example, depending on the speed of the approaching predator and other environmental cues, it might be more advantageous to freeze than to escape or vice versa (*Bhattacharyya et al., 2017*; *Evans et al., 2019*; *Yilmaz and Meister, 2013*). Moreover, it is critical for the animal to generate escapes only if 'necessary', and whether or not it is necessary to escape might be learned over time. Two-dimensional expanding shadows (looming stimuli), for example, are initially effective in triggering escapes in many settings, but with enough repetition they often lose their potency. In fact, habituation to looming stimuli has been reported in various animal species, and the speed of habituation itself is known to depend on the behavioral context (*Hayes and Saiff, 1967*; *Matheson et al., 2004*; *Oliva et al., 2007*; *Mancienne et al., 2021*; *Marquez-Legorreta et al., 2022*). However, the computational algorithms that govern interactions among sensory networks that encode distinct aspects of a looming stimulus, and the way they change in the course of habituation remains largely unknown.

It is important to recognize that a dark looming stimulus not only contains expanding motion energy, but also creates an overall dimming of the whole field in the course of its expansion. The looming and dimming aspects of a dark looming stimulus can be isolated and studied specifically in the lab using stimuli that lack the motion aspect, that is a gradual dimming of the whole field (dimming stimuli), or those that lack the dimming aspect, for example an expanding black and white checker-board pattern on a gray background that maintains the same average luminance throughout.

In larval zebrafish, neurons that are specifically tuned to looming, are largely represented in the optic tectum, specifically within the Stratum periventriculare (*Dunn et al., 2016*; *Temizer et al., 2015*). Neurons that respond to sudden or gradual changes in luminance (dimming and dark flash stimuli, respectively) are present at the level of the retinal ganglion cell arborization fields AF6 and AF8, as well as the optic tectum, pretectum, and thalamus (*Temizer et al., 2015*; *Heap et al., 2018*). It was found that the dimming aspect of a dark looming stimulus is necessary for driving escapes, as bright expanding disks rarely evoke escape, however, dimming alone is generally not sufficient (*Temizer et al., 2015*; *Mancienne et al., 2021*). In fact, *Yao et al., 2016* have identified dark flash (i.e., fast dimming) responsive dopaminergic neurons in the caudal hypothalamus that inhibit downstream escape motor networks, and are thought to help differentiate a threatening (looming) from a non-threatening (dark flash/dimming) stimulus.

To characterize the responsible underlying circuitry, we use combined behavioral experiments and 2p calcium imaging to study habituation of looming-evoked escape responses in larval zebrafish.

These experiments were executed in head-embedded tail-free fish, which is favorable for studying habituation for several reasons. First, it allows for precise stimulus control and ensures that the fish receives identical visual stimulation over repeated trials. Second, tethered fish tend to habituate faster than those freely swimming (personal observation, also see *Ahrens et al., 2012*), making it easier to achieve full habituation over shorter recording periods. Finally, this setup allows for simultaneous cellular resolution, brain wide imaging, and behavioral characterization.

Here, we show that the whole-field dimming that occurs in the course of dark looming, while incapable of eliciting escape swims, is in fact a key feature of the stimulus that drives habituation learning. Specifically, we propose that a subpopulation of dimming-sensitive (DS) potentiating neurons actively suppresses responses of the looming-sensitive (LS) neurons that relay critical information to elicit escapes in the brain of a habituating larva. We also find evidence for a separate pathway that relies exclusively on the expanding motion aspect, and that can independently contribute to the habituation process. The exact mechanism by which this second pathway may interact with looming and dimming pathways, however, is yet to be discovered.

## Results

### Looming stimuli evoke a variety of escape responses in tethered zebrafish larvae

In order to quantify the reliability and magnitude of looming-evoked escape responses in head fixed 5–7 dpf larval zebrafish we implemented the following protocol. First, larvae were embedded in agarose on a Petri dish and their tail was freed. Second, the Petri dish was placed on a platform, which was itself placed inside a larger cuboid tank on the walls of which looming stimuli were presented from the left or the right side, approximately at the height of the fish and centered on the optical axis of the eye facing that wall (*Figure 1a*, see Methods). Third, a camera was positioned above the animal, which allowed high-resolution tracking of the tail motion. Using this configuration, we found that animals executed a tail flick during the first presentation of a looming stimulus in a stochastic manner with a probability of ~60%. Measuring the rate of spontaneous tail flicks in the absence of stimulation confirmed a baseline rate of approximately 0.026 Hz on average, which did not change significantly throughout the experiment (*Figure 1—figure supplement 1a, b*). We therefore subtracted the probability of observing a spontaneous event from the overall response probability during stimulus presentation (see Methods).

Looming stimuli, with an absolute size-to-speed ratio ($l/|v|$) of 240 ms (see Methods), elicited escapes that occurred on average 1.3 s (SD = 0.5 s) before the expected collision time (*Figure 1b, c*, *Figure 1—figure supplement 1c*, negative = before collision), and these escapes displayed varying degrees of vigor, defined as the peak tail deflection amplitude (*Figure 1—figure supplement 1d*). Further, escapes were generally directed away from the side of stimulus presentation, as expected (negative amplitude = away; probability of tail flick away from the stimulus = 0.76, *Figure 1—figure supplement 1d*). Using data from these experiments, we could classify looming-evoked responses into three groups: (1) a large tail bend + a counter bend, (2) a single large tail bend, or (3) one or more small tail flicks (*Figure 1—figure supplement 2a–e*) (see Methods). For the following analysis, all response types were pooled together for simplicity, acknowledging that distinct/overlapping motor networks may be involved in triggering different response types. Pectoral fin motion and freezing responses were also analyzed in some of the experiments (*Figure 1—figure supplement 2f*, see Methods). We found that, while these specific readouts show interesting effects and correlations with the looming stimulus, they did not prove to be critical for quantification of escape responses and were therefore not included into further analysis.

### Escape behaviors habituate with repeated stimulation

We next investigated the effect of repeated exposure to dark looming stimuli on evoked escapes. To that end, we first showed a series of 10 dark looming stimuli with a specific interstimulus interval (ISI = 10 or 180 s in different sets of fish) to one eye, and then presented the same 10 stimulus sequence to the other eye, where the first eye was chosen randomly. For each of these, in total, 20 trials we determined whether, and at what time within the trial, a tail flick occurred. We used the absolute amplitude

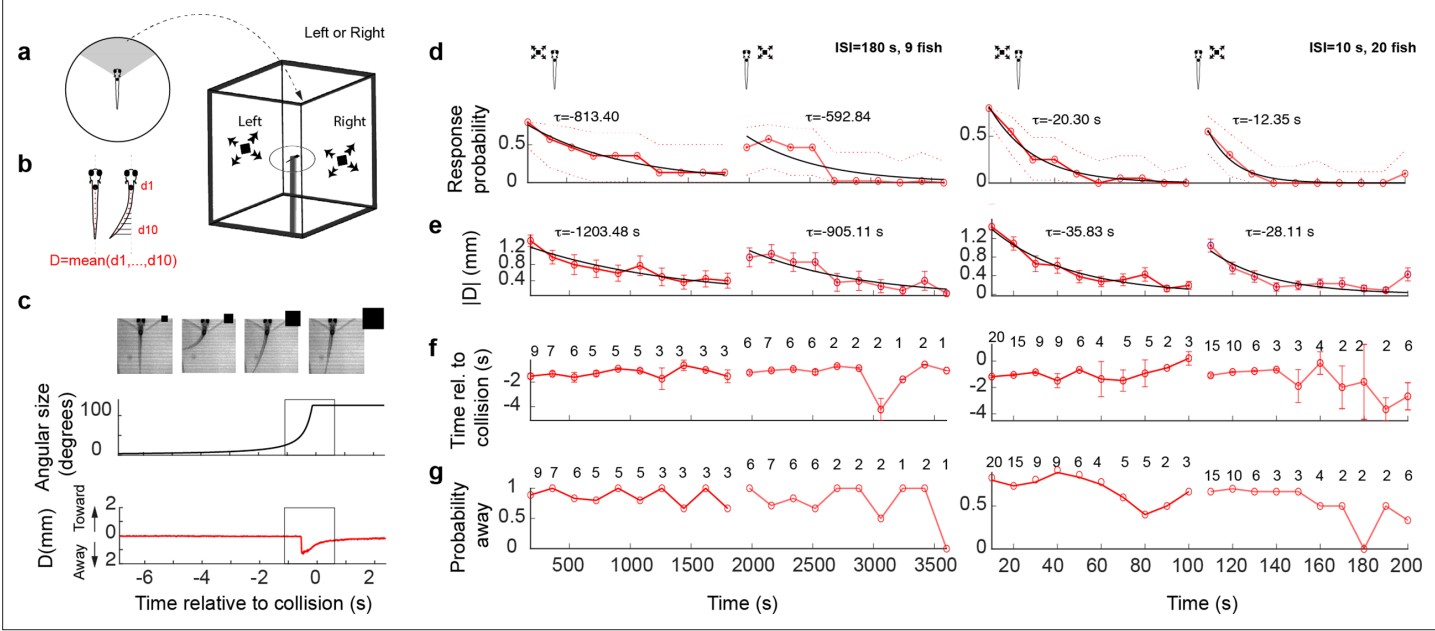

**Figure 1.** Looming stimuli evoke escapes in head-embedded tail-free larvae, which habituate in an eye- and interstimulus interval (ISI)-dependent manner. (**a**) Schematic of the experimental setup. The fish was embedded in agarose in a dish with its tail free to move and then placed in the center of a cuboid tank on the walls of which looming stimuli were presented. (**b**) The tail movement amplitude was quantified as the average displacement from the midline. (**c**) Example of a looming-evoked escape response. Four equally spaced frames taken within the gray box in the lower panels. The size of the looming stimulus shown next to the video frames is for illustration purposes only and does not reflect the actual relative size of the stimulus. The stimulus angular size and the tail flick amplitude are shown in the middle and bottom panels, respectively. (**d**) Response probability declines with stimulus repetition (red curves), with the response recovering when the stimulus is shown to the contralateral eye (light red curve to the right). The left and right panels show data from ISI = 180 and 10 s, respectively. The time for each data point corresponds to integer multiples of ISI values. $\tau$ values correspond to the time constant of the exponential fit to the data $a*\exp(-t/\tau)$. Only the very responsive zebrafish, that is those that at least responded to the first stimulus presentation were included in these experiments and the probability of observing a spontaneous tail flick within the same time window was subtracted from the observed probability. (**e**) Average amplitude of maximum peak in the tail flick trace. This includes all data points including no-escape trials where the peak amplitude was set to zero (9 fish for ISI = 180 and 20 fish for ISI = 10). (**f**) Response time relative to collision. The number of trials reflects the trials in which the escape occurred. (**g**) Percent escapes directed away from the approach.

The online version of this article includes the following figure supplement(s) for figure 1:

**Figure supplement 1.** Spontaneous tail flicks occurred at a low rate, which did not change throughout the experiment Stimulus-evoked tail-flicks were directional and phase-locked to the stimulus.

**Figure supplement 2.** Larval zebrafish generate various types of escapes and freezing responses to looming stimuli.

in the stimulus-evoked tail movement as a measure of response magnitude and the first threshold crossing of the tail movement as the timing of the response (see Methods).

We observed that response probability, as well as response amplitude, decreased with stimulus repetition in an ISI-dependent manner. The longer the ISI, the slower the rate of reduction with repetition (red curves, *Figure 1b, d, e*, left and right panels correspond to ISI = 180 and 10 s, respectively). Interestingly, the escape response largely recovered when the stimulus was switched to the contralateral eye, although a slight carry over of habituation was observed. As expected, we find that the timing of escape was tightly coupled to the stimulus onset across trials (*Figure 1f*) and that the responses were largely biased to the direction away from the stimulated eye (*Figure 1g*). For later trials, the variance increased in both values, which is likely due to habituation induced reduction in response probability.

## Dimming alone is not effective at triggering escapes, yet pre-exposure to dimming stimuli reduces responses to looming

A dark looming stimulus causes an overall reduction in the whole-field brightness in the course of its expansion. We asked whether the expansion aspect of the looming stimulus, or the whole-field dimming component alone, can evoke an escape response. To address this question, we presented

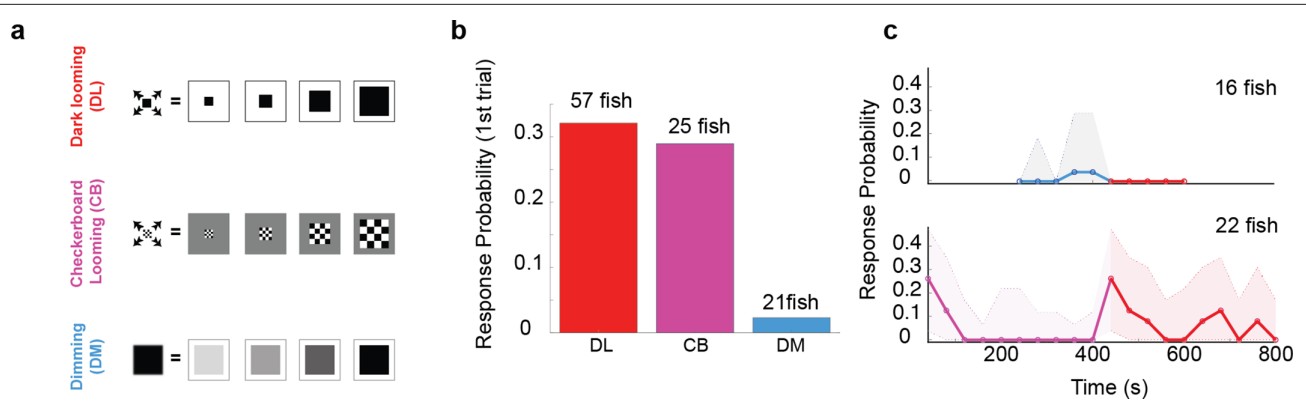

**Figure 2.** The looming aspect of the stimulus is essential for driving fast escapes, however, pre-exposure to dimming abolishes response to dark looming. (**a**) Schematics of different stimulus types used to dissect the effect of overall dimming versus expansion. (**b**) The probability of an escape response upon the first encounter with these stimulus types. A tail flick was taken as a response to the stimulus if they occurred after stimulus size had reached 5 degrees and before 5 s after the stimulus ended expanding. (**c**) Bottom panel, magenta line shows the response probability to checkerboard stimulation (10 trials, n = 22 fish), which were subsequently presented with 10 dark looming stimuli (red line). The fish attempt to escape in response to dark looming stimuli that follow 10 instances of checkerboard stimulation, but they do not respond to the same stimulus when it comes after exposure to even 5 dimming stimulations (top panel, blue and red lines: response probability to dimming and subsequent dark looming stimuli, respectively). The shaded areas show the 95% confidence interval. The probability of spontaneous flicks over the same-length time window was subtracted from the overall response probability in data presented in all panels.

larvae with three different stimulus types: (1) dark looming, which contains both dimming and expansion components, (2) 'checkerboard', which isolates the spatial expansion component by keeping the luminance at a relatively constant average level, and (3) dimming stimuli, which lack spatial expansion (*Figure 2a*). We found that both dark looming and checkerboard stimuli were effective in triggering escapes, while the dimming stimuli were not (*Figure 2b*). The amplitude and timing of escape were similar for dark looming and checkerboard looming stimuli (data not shown). Thus, the expansion aspect, and not the dimming component appears to be the critical component of a dark looming stimulus that drives fast escapes.

We next set out to assess the relative contribution of each of these components to the *habituation* process in response to dark looming. To that end, we 'pre-habituated' the fish to one stimulus type, and then tested their responsiveness to dark looming. Interestingly, we found that pre-habituation with dimming was very effective in reducing responses to the dark looming stimulus, while pre-habituation with checkerboards, that lacked the dimming aspect, had little effect.

*Figure 2c* shows the comparison between the response probability in two sets of fish. The first set, shown on the top panel, was pre-exposed to five dimming stimuli (blue line), and, remarkably, showed no response to subsequent presentation of five dark looming stimuli (red line). The second set (*Figure 2c*, bottom panel), on the other hand, was pre-exposed – and habituated completely – to 10 checkerboard stimuli (purple), and yet responded strongly to subsequent presentation of dark looming stimuli (red line). This indicates that the dimming aspect of the stimulus can be effective and sufficient for habituation to dark looming.

The significant habituation fish exhibit to checkerboard alone suggests a dimming independent pathway that likely is based on separate motion feature detectors. That is, we suggest that the 'dark looming' pathway is distinguished from this 'checkerboard' pathway by being selective for homogeneous expansion features that are absent in the expanding 'patchy' checkerboard. Thus, this pathway is protected from habituation due to repeated checkerboard exposure and explains the recovery observed in the bottom panel in *Figure 2c*.

## Looming and dimming stimuli evoke responses in distinct, as well as overlapping neuronal populations

In order to characterize the neural mechanisms underlying habituation of looming-evoked escapes, we performed 2-photon (2p) calcium imaging in individual planes of the brains of head-fixed tail-free larvae, while tracking their tail movements in response to visual stimulation. To that end, we used a

transgenic fish line (Tg(gad1b:DsRed,elavl3:H2b-GCamp7f)) that labels GABAergic neurons in red on top of a background of pan-neuronal GCamP expression, which allows the assignment of a conditional inhibitory label to every functionally identified neuron (see Methods).

Imaging data from selected planes in each fish were then mapped to the Z-brain (*Randlett et al., 2015*) and pooled across fish for subsequent brain wide volumetric analysis. In order to classify specific neuronal response types, we presented the larvae with a sequence of repeating (1) dark looming, (2) checkerboard, (3) dimming, and (4) brightening stimuli. Quantifying response profiles of individual neurons allowed us then to tease apart populations with distinct tuning properties.

For all neurons that showed response correlations with any of the four stimulus types (or for cells that correlated mostly with motor output), we calculated trial averages for each stimulus type and sorted neurons based on the strength of their average response (see Methods). Using this technique, we could divide all responsive neurons into 16 clusters (4-digit binary code corresponding to response strength to each of the four stimulus types). The four panels in *Figure 3a* show trial-averaged responses for 5384 neurons imaged in 15 fish sorted based on this 4-digit binary code (0000–1111, with 0000 corresponding to the cluster that showed weak or no sensory response).

We found that many neurons responded strongly to dimming or brightening stimuli (clusters 12 and 2, respectively), but that the next most abundant cell type consisted of neurons that were sensitive specifically to the looming stimulus (cluster 5), that is cells that robustly responded to both dark looming and checkerboard stimuli, but not to dimming or brightening. Other, smaller, populations of neurons showed varying degrees of sensitivity to dark looming, dimming or checkerboard and are presented in *Figure 3—figure supplements 1–3*. For the following analysis we will focus on the large LS and DS clusters (clusters 5 and 12, respectively). *Figure 3b* shows the average calcium activity of neurons within these clusters in response to the sequence of repeating dimming, looming, brightening, and checkerboard stimuli (five repetitions each). DS neurons respond to dark looming and dimming, and, as expected, to the sudden transition from light to dark at the end of brightening stimuli. LS neurons, on the other hand, respond to both dark looming and checkerboard looming, but not to dimming or brightening stimuli. The average responses to all stimuli tended to decline with stimulus repetition, although this decline was faster for the LS cluster. Notably, the peak activity of LS neurons occurred significantly earlier than that of DS neurons (*Figure 3b*, inset).

Interestingly, we found that neurons in the DS cluster were present in both brain hemispheres (*Figure 3c*, blue dots, hemispheric index [HI] = 0.02, with the HI of zero corresponding to equal presence in the left and right hemispheres, see Methods), whereas neurons in the LS cluster were largely localized in the hemisphere contralateral to the stimulated eye (HI = −0.86, with −1 corresponding to the exclusive presence in the contralateral hemisphere). *Figure 3d, e* shows the location of cells within LS and DS clusters mapped onto the zebrafish reference brain using the Zbrain Atlas (see Methods). Consistent with previous studies we find that LS neurons are most abundant in tegmentum, tectum (stratum periventriculare), and hindbrain networks, whereas the DS clusters dominate the pretectum, thalamus, and neuropil regions. Furthermore, we find that DS neurons were more abundant than LS neurons in forebrain regions such as thalamus, pallium, and habenula. Finally, inhibitory neurons were found to cluster with little discrimination across all regions and response types in a salt and pepper fashion (*Figure 3f*).

## Distinct subpopulation of DS neurons potentiates its response with stimulus repetition

We next examined how responses of different neuronal populations changed with stimulus repetition. To quantify variations in responses across trials, we first extracted the peak amplitudes for consecutive trials and then fitted an exponential function to these values (see Methods). This allowed us to determine the time constant for each neuron's response for each stimulus type. Based on the distribution of the time constants, we found that the majority (55%) of neurons decreased their responses and were best fit with negative time constants (*Figure 3g*, green area, see Methods). This cluster was termed 'depressing'. A smaller percentage of cells (18%) displayed an increase in response amplitude with stimulus repetition (*Figure 3g*, positive time constants, pink area), and was termed 'potentiating'. Finally, 27% of cells did not consistently change their response amplitude over repeated stimulation, and we call them 'stable'. The latter is represented by the white bar, which is centered close to zero, equivalent to a time constant of infinity.

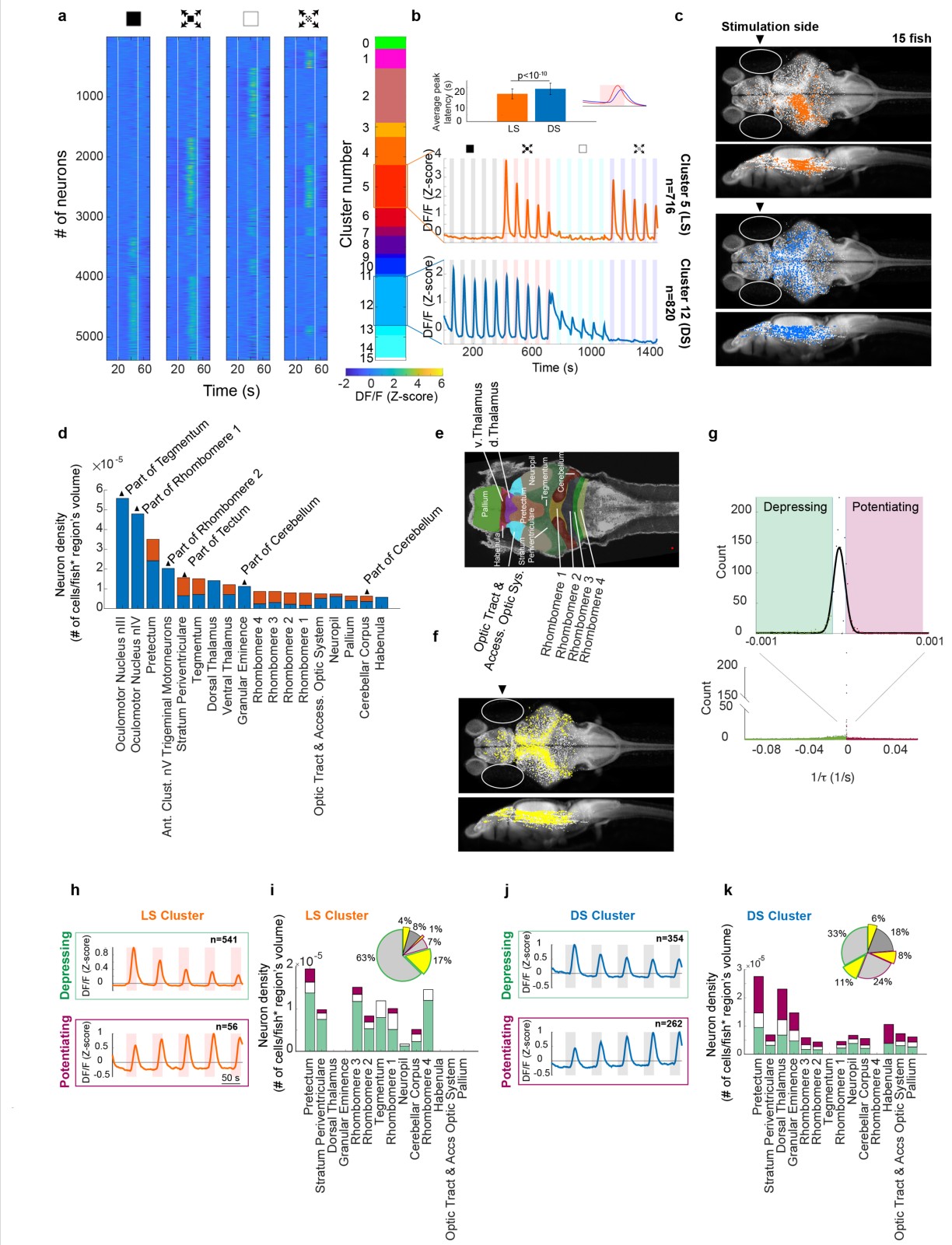

**Figure 3.** Looming stimuli evoke activity in distinct neuronal populations across brain regions, the peak response amplitudes of which could decline as well as potentiate with repeating stimulation. (**a**) First four panels show trial averages of neurons responsive to one or more of the stimulus types. Neurons were sorted based on their 4-digit binary code corresponding to higher than threshold correlation of trial averages with dimming, dark looming, brightening, or checkerboard stimuli (interstimulus interval [ISI] = 40 s, $l/|v|$ = 480 ms). The fifth panel shows the decimal cluster number

*Figure 3 continued on next page*

*Figure 3 continued*

corresponding to the binary code obtained after thresholding correlation levels of trial averages with the stimulus (see Methods). (**b**) Average (SE) response of the neurons in clusters 5 (red) and 12 (blue). Inset shows the average (SD) of response peak latency. (**c**) Distribution of looming- (LS; red) and dimming-sensitive (DS; blue) neurons in clusters 5 and 12, respectively. White dots show all neurons. (**d**) Regional distribution of all neurons in the LS (red) and DS (blue) clusters, regions are shown only if they contained at least 10 cells and were represented by at least 3 fish. (**e**) Key to the brain regions shown on the reference Z brain. (**f**) Distribution of putative GABAergic neurons (yellow) among all neurons (white). (**g**) Histogram of the coefficients of the exponential fit function to the peak response amplitudes of all neurons (see Methods). Full-width-half-max of a Gaussian fit to the near zero peak was used to determine positive ($4.13 \times 10^{-5}$ s$^{-1}$) and negative ($-1.17 \times 10^{-5}$ s$^{-1}$) thresholds. Maroon and green segments correspond to potentiating and depressing groups, respectively. (**h**) Median responses of depressing and potentiating LS cell subclusters to the first five dark looming stimuli. (**i**) Brain region stacked histogram of depressing (green), potentiating (maroon), and stable (white) LS cells. Inset: the proportion of GABAergic neurons (yellow) within each subcluster (border colors corresponding to depressing, potentiating, and stable subclusters, gray: non-GABAergic) (**j, k**). Same as (**h, i**) for DS cells.

The online version of this article includes the following figure supplement(s) for figure 3:

**Figure supplement 1.** Anatomical location and response properties of identified neuronal clusters.

**Figure supplement 2.** Anatomical location and response properties of identified neuronal clusters.

**Figure supplement 3.** Anatomical location and response properties of identified neuronal clusters.

When we applied this classification to the LS and DS clusters, we found that the great majority of LS neurons belong to the depressing group (*Figure 3h*, top panel, *Figure 3i*, green segments), and that their potentiating fraction was much smaller (*Figure 3h*, bottom panel, *Figure 3i*, maroon segments). The DS cluster on the other hand contained a prominent subcluster of potentiating cells (DS$_{POT}$, *Figure 3j, k*). Interestingly, *Figure 3k* shows that DS$_{POT}$ neurons (maroon segments) present a significant portion of neurons in a large fraction of the brain regions we recorded from, including areas such as the habenula and the palium which are implicated in the adaptive and dynamic control of behavior.

The prominence of DS$_{POT}$ neurons, combined with the finding that pre-habituation with dimming stimuli strongly reduces responsiveness to looming (*Figure 2c*, top panel), suggests a possible model where these potentiating DS cells exert a pivotal role in silencing a hardwired escape circuit that relays looming-specific visual signals to downstream motor regions. Such an inhibition could either target the LS neurons directly, as suggested by the dominant fraction of declining LS neurons in the tectum, pretectum, and tegmentum, or, alternatively, such inhibitory DS neurons could target-specific centers in the hindbrain, such as rhombomeres 1–3, that control escape behavior (see *Figure 3k*). Importantly, this interpretation makes a strong prediction that a significant fraction of the DS$_{POT}$ neurons should be inhibitory in nature. To directly test this hypothesis we made use of the GABAergic label in our fish line (see *Figure 3f*), and directly quantified the proportion of inhibitory neurons in all of our cell classes. Consistent with the proposed model, we find that the DS$_{POT}$ cluster contained more than 25% neurons with a GABAergic label, whereas the fraction of GABAergic neurons in the LS and declining DS clusters was 10% or less (*Figure 3i, k*, insets).

We next set out to test whether these DS$_{POT}$ inhibitory neurons indeed play a critical role in suppressing LS activity, which in turn is likely responsible for driving the escape behavior (*Figure 2*). To that end, we leveraged the fact that response strength in DS$_{POT}$ neurons is under experimental control and can be enhanced by repeated dimming stimulation. Specifically, we measured the response size in LS neurons either after an exposure to 10 dimming stimuli, or after an equivalent waiting time with no stimulation (*Figure 4a*, see Methods). Importantly, LS neurons are similarly silent during the dimming stimulation period, as they are during the waiting period (*Figure 4b, c*). We find that such 'covert' potentiation of the dimming pathway led to a significantly reduced response size in LS neurons when compared to the largely – but not completely – recovered response size after a 10-min waiting period (*Figure 4d*).

In summary, we propose that DS$_{POT}$ neurons play a critical role in the process of habituation to dark looming stimuli. Specifically, we believe that they do so by an incremental increase of their inhibitory influence onto LS neurons, which themselves relay the information necessary to elicit escapes.

## Degree of binocularity in LS and DS neurons

We showed that LS neurons were largely confined to the contralateral hemisphere of the stimulated eye, whereas DS neurons could be present in either hemisphere (*Figure 3c*, blue cluster). The fact

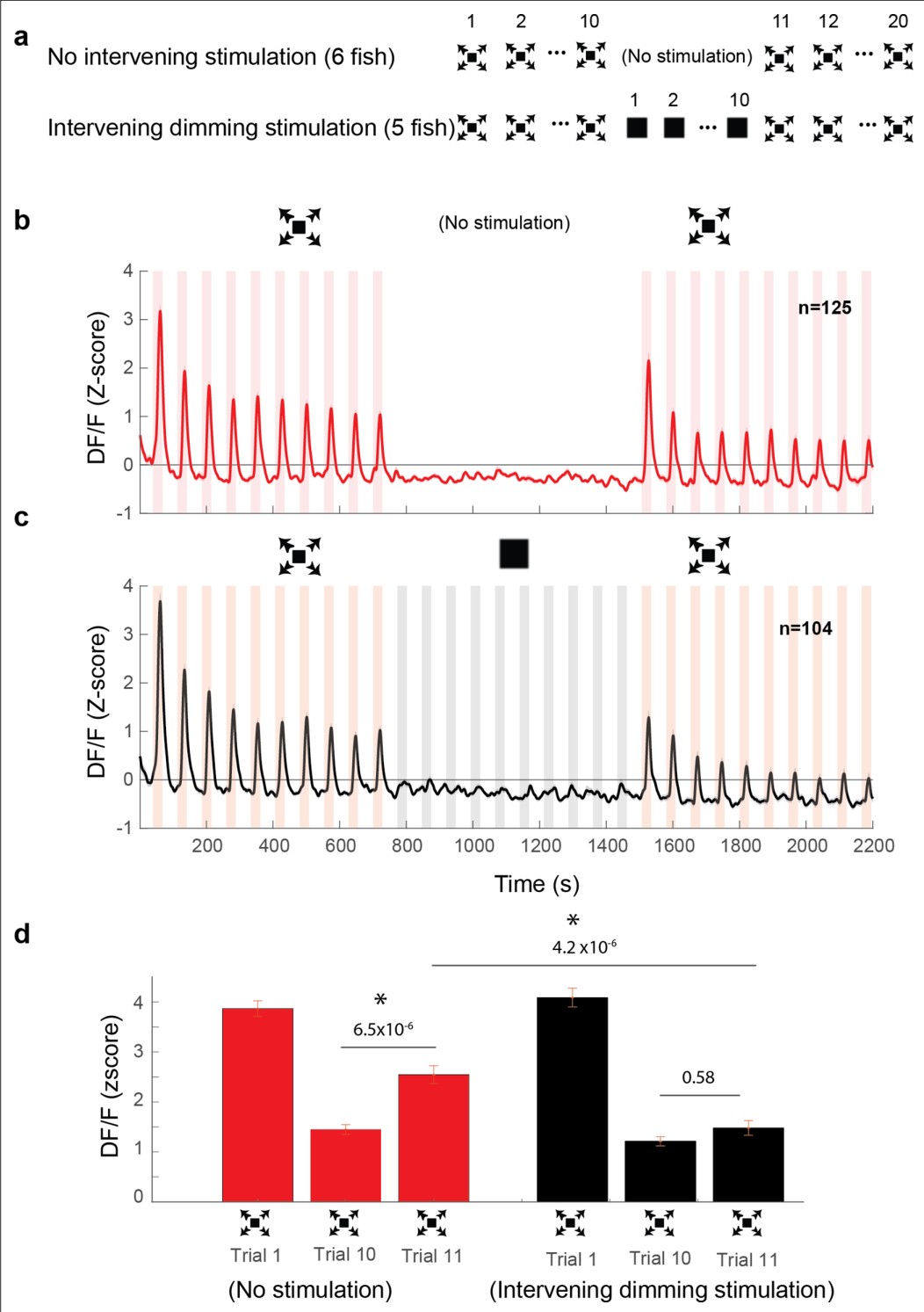

**Figure 4.** Dimming stimuli negatively affect response recovery of looming-specific neurons. (**a**) Stimulation paradigms for the two experimental groups: one with dimming stimuli presented in between two dark looming sequences and the other with no stimulation for an equal period of time. (**b**) Average response of looming responsive decreasing neurons (interstimulus interval [ISI] = 40 s). Dark looming stimuli were presented, followed by a recovery period (equal to the timing of 10 stimulus presentations at ISI = 40 s), after which dark looming stimuli were presented again. (**c**) Same as (**b**) except that instead of the period with no stimulation, the fish was stimulated with dimming stimuli. (**d**) The peak response to dark looming was significantly smaller on the 10th compared to the first trial as expected. This response has significantly recovered on the first dark looming test trial

*Figure 4 continued on next page*

*Figure 4 continued*

after waiting period (Trial 11), but not in the case of intervening dimming stimulation. The peak response on the 11th trial for the dimming group was significantly smaller than that for the wait group. Error bars = standard error. * indicates statistical significance.

that all retinofugal projections in the zebrafish are exclusively contralateral indicates that LS neurons must be largely monocular. DS neurons on the other hand are distributed across both hemispheres and therefore no such conclusions can be made based solely on experiments where only one eye is stimulated (*Figure 3*). Therefore, we explicitly tested the binocularity of each neuron by stimulating first one eye and then the other, and we then compared the cells' responses across the two stimuli. We found that, as predicted, LS neurons responded mainly to stimuli presented to the contralateral eye (*Figure 5a*, HI = −0.89 and −0.94 for right and left eye stimulation, respectively, see Methods), whereas DS neurons come in two classes: one class consists of monocular cells which are largely located on the contralateral side (*Figure 5b*, HI = −0.52 and −0.47 for right and left eye stimulation, respectively), and a second class that consists of binocular neurons which are evenly distributed across hemispheres (*Figure 5c*, HI = −0.04, see Methods).

## DS neurons' activity exhibits various levels of interocular transfer

We next focused on the two DS clusters (monocular and binocular), and asked about the respective proportion of potentiating neurons ($DS_{POT}$) in each. *Figure 6a* shows that in both the binocular and the monocular populations of DS neurons, about one-third of the individuals belong to the potentiating class. Average traces of these potentiating classes are shown in *Figure 6b, c*. The general lack of transfer in behavioral habituation between the two eyes (*Figure 1d and e*) argues that $DS_{POT}$ neurons exert their inhibitory influence in an eye-specific manner which suggests that the monocular $DS_{POT}$ neurons are the critical components for this feature. In other words, we propose that the two non-overlapping classes of $DS_{POT}$ neurons, that is the ones that respond to either the left or the right eye, control habituation in the left and right eyes, respectively (*Figures 5b and 6b*). The residual transfer in habituation that we observe in behavioral experiments predicts the existence of binocular $DS_{POT}$ neurons that continue their trend of potentiation when the stimulus is switched across eyes (*Figure 6c*, top panel). Note that the number of these cells (*n* = 46) is smaller than the total number of monocular

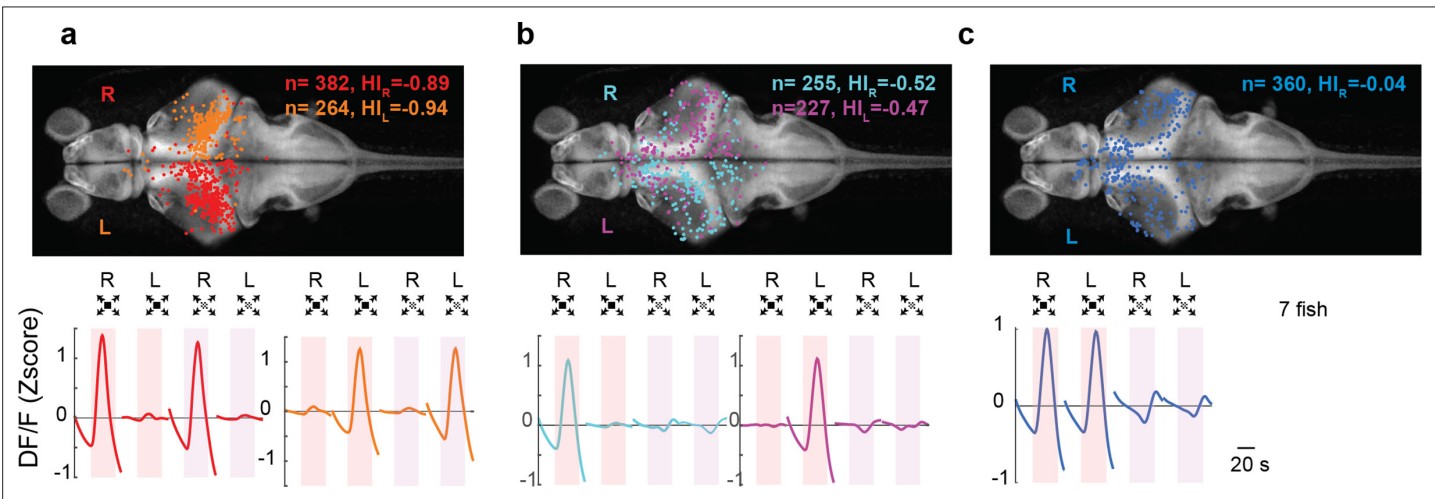

**Figure 5.** Looming-sensitive neurons were monocular, whereas dimming-sensitive neurons could be binocular or monocular. (**a**) Looming-sensitive neurons respond to both dark looming and checkerboard stimuli, but only when they are presented to the eye contralateral to their location in the brain. Top panel shows the location of the cell bodies that respond to right eye (red) and left eye (orange) stimulation. The bottom panel shows trial-averaged responses of these neurons. HI: hemispheric index, negative values indicate bias toward the hemisphere contralateral to the stimulated eye. (**b**) Dimming-sensitive neurons that responded to stimulation of one or the other eye. These neurons were located largely in the hemisphere contralateral to the eye they responded to, although the bias was not as large as that in looming-sensitive neurons. (**c**) Dimming-sensitive neurons that responded to the stimulation of either eye were present in both contralateral and ipsilateral hemispheres. Data from seven fish, *n* indicates the number of neurons.

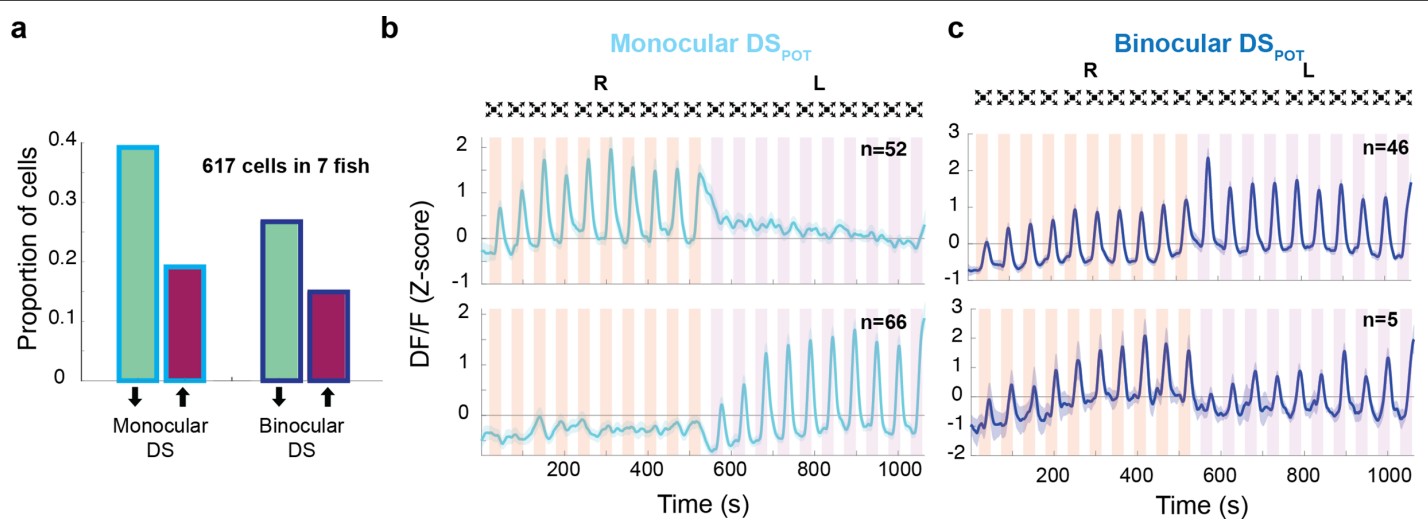

**Figure 6.** Response properties of binocular and monocular dimming-sensitive neurons. (**a**) Proportion of dimming-sensitive neurons that fall into binocular/monocular categories and further into potentiating (maroon) and depressing (green). (**b**) Top and bottom panels show the average of monocular DS_POT neurons responsive to the right and left eyes, respectively (interstimulus interval [ISI] = 20 s). (**c**) Top panel shows the average response of binocular DS_POT neurons that retain or continue increasing their amplitude on the second side. Bottom panel shows the few binocular DS_POT neurons that reset their response when the stimulus was switched to the contralateral side.

DS$_{POT}$ cells ($n$ = 118). We also found a small number of binocular resetting DS$_{POT}$ neurons (*Figure 6c*, bottom panel, $n$ = 5) who might contribute to the monocular specificity of the habituation process.

In summary, these findings allow us to propose a model of habituation, where an inhibitory subpopulation of monocular DS neurons potentiate their response amplitude when the animal is exposed to repeated looming stimuli. We further propose that this inhibition acts largely locally and within an eye-specific pathway, and thereby prevents activity to propagate through an excitatory and hardwired LS network that ultimately elicits escape swims in the fish.

## Discussion

In this study, we used a tethered behavioral setup (*Figure 1a, b*) and 2p calcium imaging to investigate circuit mechanisms that underlie habituation learning in larval zebrafish. Consistent with studies in freely swimming larvae (*Mancienne et al., 2021*; *Marquez-Legorreta et al., 2022*) we found that the tethered larvae habituate to repeated stimulation with dark looming stimuli presented from the side to one eye, in an ISI-dependent manner. Additionally, we show that the escape response largely recovers when the stimulus is switched from one eye to the other (*Figure 1d, e*).

Dark looming stimuli not only contain an expanding edge but also cause a reduction in whole-field luminance in the course of their expansion. Interestingly, we found that pre-exposure to just the dimming component habituates responsiveness to dark looming in a comparable fashion than repeated exposure to the full dark loom itself (*Figure 2c*). Pre-exposure to checkerboard stimuli on the other hand, which lack this global dimming aspect, did not induce a significant reduction when probed with a dark looming stimulus, in spite of the prominent habituation observed in response to the checkerboard itself. This suggests that there are two, largely independent, habituation pathways at work. The one described here acts through an inhibitory population of DS neurons that potentiate upon repeated exposure to a dimming stimulus. A second mechanism appears to act exclusively within the checkerboard pathway and the habituation process is likely explained by an activity-dependent mechanism that is intrinsic to that feedforward pathway. The preferred approach to uncover such mechanisms requires molecular and cell biological tools (*Randlett et al., 2019*) which is beyond the scope of our study.

Our results differ from reports from a previous study where looming stimuli were presented from below, rather than from the side as in our experiments, and where animals were freely swimming and not tethered. In this study, pre-exposure to dimming did not seem to significantly affect the response

to subsequent dark looming (*Mancienne et al., 2021*), which could be explained by the different experimental approach.

In order to shed more light on the putative role of the dimming pathway in habituation, we investigated the neural basis of this behavior using 2p calcium imaging, and we quantified brain activity within regions where looming-responsive neurons were previously reported (*Dunn et al., 2016*). This allowed us to identify two prominent functional clusters of neurons: the first is LS and responds primarily to the expanding motion component, the second is DS, it responds primarily to whole-field dimming, and is largely insensitive to motion. This is consistent with previous findings that reported prominent functional response clusters that are either sensitive to dark looming or to dark flash, which is in essence a very fast dimming stimulus, in the midbrain of larval zebrafish (*Dunn et al., 2016*; *Temizer et al., 2015*; *Heap et al., 2018*).

Interestingly, we found that although LS neurons were mostly confined to the brain hemisphere contralateral to the stimulated eye, dimming responsive neurons could be found in either hemisphere (*Figure 3c*). Further, we find that, while most cells tended to decrease in responsiveness with repeated stimulation, as reported in previous studies (*Mancienne et al., 2021*; *Marquez-Legorreta et al., 2022*), a significant subpopulation of DS neurons potentiated their responsiveness. These findings allow us to propose a conceptual circuit model where these potentiating dimming-sensitive neurons (DS$_{POT}$) contain a significant fraction of inhibitory cells that play an explicit role in suppressing behavioral responses by incrementally silencing the brain stem escape network, particularly through the population of LS neurons that relays looming information to motor output structures. In order to validate this conceptual model, we tested whether DS$_{POT}$ neurons are capable of covertly depressing LS neurons in the tectum, and we found that, indeed, pre-exposing the fish to repeated dimming stimulation significantly depresses the amplitude of LS neurons' responses to dark looming stimuli (*Figure 4*).

The prominent recovery of behavioral responses when a stimulus is presented to the 'un-habituated' eye (*Figure 1*), combined with the striking difference between the eye-specific anatomical distribution of LS and DS clusters, allowed us to further constrain our conceptual circuit model. While LS neurons were found to be largely monocular and located only on the contralateral hemisphere, DS neurons came in two flavors: monocular and contralaterally placed on the one hand, and binocularly responsive neurons that were distributed across both hemispheres, where both groups contained depressing as well as potentiating cells (*Figure 6b, c*).

The prominent presence of monocular DS$_{POT}$ neurons is consistent with a model where monocular DS$_{POT}$ neurons exert local inhibition onto the monocular LS neurons, an effect that is lifted once the stimulus is switched to the contralateral eye, where the local DS$_{POT}$ have not yet been potentiated. The presence of a smaller number of binocular DS$_{POT}$ neurons, who will undergo potentiation during the stimulation of either eye, could explain the residual transfer of habituation between eyes, and provides additional validation of the local inhibition model. Such binocular DS neurons were recently also reported in the larval zebrafish tectum (*Tesmer et al., 2022*) where the pathway of interhemispheric transfer was shown to be located in the Torus Longitudinalis, which itself receives input from the tectum (*DeMarco et al., 2020*). However, we do not have conclusive evidence for an anatomical connection between DS$_{POT}$ and LS neurons. Targeted ablation of DS$_{POT}$ neurons could potentially test the necessity of this population for habituation, but their distributed nature makes it unlikely that a sufficiently large fraction could be targeted to expect a significant effect. Alternatively, anatomical connectivity could be demonstrated directly by either viral tracing or EM-based connectomics approaches, but such technologies are currently not mature enough to be readily deployed in our study.

In addition to the DS$_{POT}$ neurons located in the tectum, which play a pivotal role in looming habituation in our model, we have located DS$_{POT}$ neurons with significant densities in various other regions of the larval zebrafish brain, such as the pretectum, dorsal thalamus, and habenula. We suggest that these neural populations might become relevant in different behavioral contexts such as associative learning and experience-dependent changes in fear response (*Amo et al., 2014*; *Agetsuma et al., 2010*; *Palumbo et al., 2020*), as has been shown for the habenula in adult fish (*Amo et al., 2014*), but also other organisms including primates (*Dayan and Huys, 2009*; *Nakamura et al., 2008*) in various other behavioral assays.

In addition to modulating response strength in these regions, DS$_{POT}$ neurons are also well positioned to inhibit hindbrain motor centers directly. Indeed, we show that pre-habituation with dimming

stimuli silences behavioral output to dark looming (*Figure 2*), but leaves LS neurons in the tectum still responsive (*Figures 3b and 4c*), which suggests that the behavioral suppression can to some extent bypass the tectum and act directly on hindbrain motor regions.

This hypothesis is supported by previous studies that have shown that axons of both looming and dimming responsive neurons project to hindbrain motor regions (*Helmbrecht et al., 2018*; *Zottoli et al., 1987*; *Sato et al., 2007*). Interestingly, a group of dark flash responding dopaminergic neurons was identified in the caudal hypothalamus and was shown to drive inhibition of the Mauthner cells via hindbrain glycinergic interneurons (*Yao et al., 2016*). This dopaminergic population offers an attractive and plausible target for the $DS_{POT}$ neurons described here, which could thus confer an increase in inhibition to the Mauthner cells through the same pathway. Indeed, glycinergic inhibition is thought to play a critical role in modulating responsiveness of hindbrain escape circuitry in general (*Koyama et al., 2011*), and it has been shown to be specifically implicated in dampening responsiveness of Mauthner cells during habituation of acoustic startle response (*Marsden and Granato, 2015*). Thus, the details of how $DS_{POT}$ neurons modulate hindbrain motor networks, and the precise mechanisms by which they might specifically act on the Mauthner escape network, is an important area for future research.

In summary, our data allow us to propose a realistic circuit model where a subset of inhibitory DS neurons are incrementally potentiated by repetitive stimulation and where they serve to locally depress the looming selective relay pathway within tectal circuitry. We further suggest that these DS neurons act within one hemisphere and in an eye-specific fashion. As such, this model generates a series of testable predictions about behavior, neural response properties and synaptic connectivity which are all eminently testable and straightforward to validate – or invalidate – in future experiments.

## Methods

### Experimental setup and stimulus presentation

All experiments followed institution IACUC protocols as determined by the Harvard University Faculty of Arts and Sciences standing committee on the use of animals in research and teaching (IACUC 22-04). Fish were embedded in agarose on a 35-mm Petri dish and their tail and pectoral fins were freed (*Figure 1a*). The dish was in turn placed on a platform inside and in the center of a larger (170 $H \times 130$ $W \times 75$ $D$, mm) rectangular tank, on the walls of which the looming stimulus was presented (*Figure 1b*). The fish were allowed to acclimate for about 30 min before presentation of looming stimuli. Their movement was filmed at 120 Hz. The movement of the tail was quantified frame-by-frame as the average deflection of 10 equidistant points along the tail from the midline (*Figure 1b, c*). A tail flick was detected when the average tail deflection exceeded a threshold.

In these experiments (data shown in *Figure 1d* – only), fish with low response probability (defined as those that did not respond to the first stimulus presentations) were not tested further and were not included in the analysis. In this case, therefore, the observed probability for the first trial, before correction for spontaneous flick rate (see behavioral data analysis bellow), was 1.

Looming stimuli were simulated using MATLAB and presented at 20 Hz refresh rate to one eye or the other, 10 times each at varying ISIs. The screen on the unstimulated side was kept dark (whole-field black projected on the tank wall on that side) the whole time when the other side was being stimulated. For the data shown in *Figure 1*, the stimulus size to speed ratio ($l/|v|$) was chosen at 240 ms with the initial and final angular size equal to 4 and 140 degrees, respectively. The value of $l/|v|$ was found effective in triggering escapes after a set pilot of pilot experiments. The ISI values chosen for these experiments were 10 and 180 s. At the end of the approach sequence, the stimulus remained at its final size for 2.5 s after which it disappeared for the duration of ISI. Behavioral experiments were initially performed in either open or closed loop. In closed loop experiments, looming was stopped as soon as the fish generated an escape response, whereas in open loop experiments it continued expanding to its full final size regardless of the fish's behavior. In closed loop experiments, the stimulus size was kept at the stopped value for the remainder of the time where it would have expanded had the escape not happened. There was no significant difference in response probability and timing for the two conditions (data not shown), and their data were pooled. All experiments under 2p calcium imaging were performed in the open loop configuration.

For calcium imaging experiments, a similar setup was used, except that the tank was smaller (60 $H$ × 55 $W$ × 55 $D$, mm), and the fish was embedded in agarose on a 20 × 20 mm agarose-coated glass coverslip mounted on a platform such that it was centered on the stimulus shown on the tank wall. In these experiments, all fish were retained for the analysis regardless of their responsiveness to the stimulus. This could also give us an estimate of the probability of responsiveness to the first stimulus presentation across fish (see *Behavioral data analysis*, *Figure 3b*). Four different stimulus types were presented repeatedly in various orders: dark looming (DL), checkerboard looming (CB), dimming (DM), and brightening (BR). The time course of the dimming stimulus was matched to the dimming profile of the dark looming stimulus calculated based on the proportion of the dark versus bright pixels. Brightening stimulus had the same time course as dimming except for the inverse polarity. Note that whole-field brightening also occurs after each dark looming and dimming stimuli go back to their initial size after having reached their final size. In these experiments, the stimulus $l/|v|$ was set to 480 ms and the initial and final sizes were 3 and 80 degrees, respectively. The stimulus remained at its final size for 10 s after which it disappeared for the duration of ISI. The projector refresh rate was 30 Hz and the frame rate for the behavioral monitoring camera was 25 Hz.

## Behavioral data analysis

Video recordings were analyzed to extract the position of the tail and in some experiments the pectoral fins at each frame. The response amplitude was quantified as the average deflection of each point along the tail from the midline. Escape response time was selected as the timing of the amplitude crossing the threshold (10 pixels or 0.2 mm) prior to the first peak, which had a threshold of 20 pixels or 0.4 mm. In some trials, tail flicks occurred too early or too late to be considered as stimulus evoked based on previous literature. This amounted to less than 3% of the trials which were excluded from further analysis. The criterion for exclusion was if the responses occurred before the stimulus had reached 5 degrees in angular size, and those that occurred more than a second after projected collision time. The response probability for each trial was calculated as the percentage of fish that generated a tail flick within each trial. In order to account for spontaneous, non-stimulus-evoked, tail flicks, we subtracted the probability of observing a spontaneous response (i.e., probability of observing one or more spontaneous tail flicks) within the same duration, assuming a Poisson distribution for spontaneous flicks occurring. *Figure 1—figure supplement 1* shows the spontaneous flick rate for an exemplar experimental group where the fish was presented with 10 dark looming stimuli in a row. The spontaneous rate varied across experiments/animals and had a mean (SD) of 0.0266 (0.0730) Hz. Average spontaneous rates were calculated separately for each experimental group and were subtracted from the observed probabilities in those groups. Response amplitude was selected as the largest peak amplitude in the course of the approach sequence. Fish's response type was further divided into five categories (*Figure 1—figure supplement 2a–c*). Response time, amplitude of the positive peak and/or negative peak, and their ratio were used together with $K$-means algorithm to classify response types (*Figure 1—figure supplement 2c*). In some fish, we freed the pectoral fins from agarose to quantify their movement in response to the stimulus. Some of those fish showed spontaneous periodic pectoral fin movements (*Figure 1—figure supplement 2b*, right column). These movements were tracked in a small window around the left and right fins using the standard deviation of the difference between the brightness value of consecutive frames calculated for each fin. The deviation traces for left and right fin movement were then averaged, smoothed, and used to calculate the timing of peak fin movement and the fin beat frequency and timing. Consistent with prior studies (*McClenahan et al., 2012*), we found that the beating of the fins stopped that is they were 'tucked-in' before large tail flicks (*Figure 1—figure supplement 2b*, right column and *Figure 1—figure supplement 2f*). We further observed that in some trials, the fins stopped beating even in the absence of a tail flick (*Figure 1—figure supplement 2a, c*, iv). Such behaviors are likely a form of freezing response. Interestingly, the timing of fin tuck in freeze or escape trials did not show a significant difference (*Figure 1—figure supplement 2f*), indicating that freezing might be a 'halted' escape response.

## Calcium imaging experiments

Transgenic fish that expressed calcium indicators Tg(elavl3:H2b-GCamp7f) pan-neuronally were used in all experiments. In some experiments, double transgenic larvae that expressed a red label in their

GABAergic neurons were used Tg(gad1b:DsRed,elavl3:H2b-GCamp7f). Images were acquired at 1 Hz at 1024 × 1034 pixels resolution. Only one plane was imaged per fish in order to characterize the response dynamics of cells from the first encounter to a given stimulus. For each fish, an anatomy stack was acquired and used to map the imaging plane to the Zbrain Atlas (*Randlett et al., 2015*). CaImAn software (*Giovannucci et al., 2019*) was used to extract fluorescent data and Advanced Normalization Tools (ANTS) method was used to first map the anatomical stacks acquired from each fish to the reference brain. The same mapping parameters were then used to map the location of units identified by CaImAn software to the reference brain. Data were then pooled across fish before further analysis. Imaging planes were chosen in regions where looming and dimming responsive neurons were previously reported (*Dunn et al., 2016*), excluding deep and superficial planes, and were focused around the optic tectum (*Figure 3*, *Dunn et al., 2016*). Units that showed activity correlated to the overall stimulus sequence, ISI, or motor response were extracted from this data and analyzed further. The correlation threshold was selected empirically as 0.3 for stimulus and ISI, and 0.5 for motor response. In the subset of double transgenic fish that expressed a red label in their GABAergic neurons, a series of scans were performed at the end of the experiment, and an average image was calculated using the red channel. This image was binarized and overlaid on the Region Of Interest (ROI) of active cells as identified by CaImAn. A cell was considered GABAergic if the red label covered more than 70% of its ROI.

## Clustering of cells based on their tuning properties

For identifying responsiveness to each stimulus type, the average response of each cell to each stimulus type was calculated across repetitions. The cell was considered responsive to a given stimulus type if the correlation of its average response profile (across stimulus types) showed a correlation higher than 0.2 with the given stimulus regressor. For experiments shown in *Figure 4*, LS and DS clusters were identified based on their relative responsiveness to checkerboard and dark looming stimuli. Checkerboard stimuli were presented before the start of the test sequence for these experiments which was dark loom, dim, dark loom or dark loom, wait, dark loom.

## Analysis of calcium response dynamics during habituation

An exponential fit, $a \exp(t/\tau) + c$, was used to quantify the dynamics of response peak amplitudes to dimming dark looming and checkerboard stimuli with trial repetition, wher $\tau$ is the time constant and $t$ is the timing of each peak. The fit was only performed on cells that responded to at least three repetitions of the stimulus. In some cases, the exponential function was not a good fit; these were identified based on the large confidence intervals around the time constant (10 times or larger than the largest exponent; $1/\tau$). Nevertheless, these fits captured correctly the overall trend of amplitude change (increasing or decreasing). These time constants were not included for calculating positive and negative threshold values (as described below, not included in *Figure 3g*), but once these thresholds were calculated, these neurons were assigned to the potentiating and depressing groups based on the sign of their exponent. To calculate positive and negative thresholds, the histogram of well-fit exponents ($1/\tau$), which exhibited a large peak near zero and another prominent peak at more negative values was fit with a double Gaussian function. The full-width-half-max of the near zero Gaussian was then used to determine positive and negative thresholds for the $1/\tau$ variable. The center Gaussian mean $= -3.76 \times 10^{-5}$ (95% confidence interval $-3.87 \times 10^{-5}$, $-3.66 \times 10^{-5}$) and the left Gaussian mean $= -0.0075$ (95% confidence interval $-0.0085$, $-0.0065$) were both significantly smaller than zero.

## Quantifying binocularity and hemispheric indices of LS and DS neurons

To assess the binocularity of LS and DS clusters, we presented 10 dark looming stimuli to one eye and then 10 to the next, followed by 10 checkerboard looming presented to the first eye, and then 10 checkerboard looms to the second eye and quantified tuning of neurons within those clusters based on their trial averages. We then classified LS neurons as those that responded to both dark checkerboard looms and DS neurons as those that responded to dark looms but not checkerboard looms (*Figures 5 and 6*).

In the single eye stimulation experiments shown in *Figure 3*, the hemispheric indices were calculated using the following formula:

HI = (difference between the number of cells in the hemisphere ipsilateral and contralateral to the stimulus)/total number of cells.

A negative HI indicates a bias toward the contralateral side.

For the binocular stimulation experiments, the right and left hemispheric indices ($HI_R$ and $HI_L$) for monocular DS and LS neurons were similarly defined.

$HI_{R(L)}$ = (difference between the number of cells selective to the right (left) eye in the right (left) and left (right) hemisphere)/total number of cells.

The HI for binocular dimming responsive neurons was calculated relative to the right eye as follows:

$HI_R$=(Difference in the number of cells in the hemisphere ipsilateral and contralateral to the right eye)/total number of cells.

### Custom analysis software and statistics

MATLAB (MathWorks) software was used for all analysis and stimulus presentations. Data analysis software can be accessed on GitHub (https://github.com/halehfoto/Fotowat_Engert_eLife_Code.git; *Fotowat, 2023*). All p values reported are calculated using non-parametric Kruskal–Wallis test.

## Acknowledgements

We would like to thank Sumit K Vohra for his help with brain registration.

Florian Engert received funding from the National Institutes of Health (U19NS104653, 1R01NS124017), the National Science Foundation (IIS- 1912293), and the Simons Foundation (SCGB 542973). Haleh Fotowat received funding from the Swartz Foundation Postdoctoral Fellowship.

## Additional information

### Funding

| Funder | Grant reference number | Author |
|---|---|---|
| National Institutes of Health | 1R01NS124017 | Florian Engert |
| Simons Foundation | SCGB 542973 | Florian Engert |
| National Science Foundation | IIS- 1912293 | Florian Engert |
| National Institutes of Health | U19NS104653 | Florian Engert |

The funders had no role in study design, data collection, and interpretation, or the decision to submit the work for publication.

### Author contributions

Haleh Fotowat, Conceptualization, Data curation, Software, Formal analysis, Validation, Investigation, Visualization, Methodology, Writing - original draft, Writing - review and editing; Florian Engert, Conceptualization, Resources, Supervision, Funding acquisition, Methodology, Writing - original draft, Project administration, Writing - review and editing

### Author ORCIDs

Haleh Fotowat http://orcid.org/0000-0003-0372-4912
Florian Engert http://orcid.org/0000-0001-8169-2990

### Ethics

All experiments followed institution IACUC protocols as determined by the Harvard University Faculty of Arts and Sciences standing committee on the use of animals in research and teaching (IACUC 22-04).

**Decision letter and Author response**
Decision letter https://doi.org/10.7554/eLife.82916.sa1
Author response https://doi.org/10.7554/eLife.82916.sa2

## Additional files

### Supplementary files
• Transparent reporting form

### Data availability

The experimental datasets generated in this study and used for making plots shown in Figures 1–6 can be accessed through Harvard University's data repository system https://doi.org/10.7910/DVN/EWQOJB. The files are organized into folders named after the figure numbers. The file named 'Code List_Data Info - eLife.pdf' contains information about the collection of custom MATLAB codes that were used to generate the plots, which can be accessed through GitHub at https://github.com/halehfoto/Fotowat_Engert_eLife_Code (copy archived at *Fotowat, 2023*).

The following dataset was generated:

| Author(s) | Year | Dataset title | Dataset URL | Database and Identifier |
|---|---|---|---|---|
| Fotowat H | 2023 | Data for Neural Circuits Underlying Habituation of Visually Evoked Escape Behaviors in Larval Zebrafish | https://doi.org/10.7910/DVN/EWQOJB | Harvard Dataverse, 10.7910/DVN/EWQOJB |

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
