## [Editor Report]

This study presents a valuable finding on how visual stimuli are processed during the habituation of visually evoked escape behaviors. The evidence supporting the claims of the authors are convincing, although future study to show the connectivity and function of the neural circuits would be important to get more insights into this process. The work will be of interest to neuroscientists working on habituation and learning.

---

## [Decision Letter]

**Decision letter after peer review:**

Thank you for submitting your article "Neural circuits underlying habituation of visually evoked escape behaviors in larval zebrafish" for consideration by *eLife*. Your article has been reviewed by 2 peer reviewers, and the evaluation has been overseen by a Reviewing Editor and Tirin Moore as the Senior Editor. The reviewers have opted to remain anonymous.

Essential revisions:

The strength of the present work is that the authors discovered a subpopulation of inhibitory neurons, that is potentiated during repetitive visual stimuli and control the habituation of visually evoked escape behavior. This is a great finding and should be shared with the community. I see the following weakness in the manuscript.

1) The lack of any further experiments on "checkerboard stimuli", (2) no connectivity between DS and LS was shown.

a) The authors emphasize the existence of another pathway for habituation induced by "checkerboard stimuli", but this has not been characterized further.

2) Although the authors discovered the DS neurons, which are potentiated during repeated stimuli and inhibit LS neurons, the actual connectivity has not been demonstrated.

We would like you to respond to the reviewer's comments below.

*Reviewer #1 (Recommendations for the authors):*

Specific comments:

1) Results, the first paragraph, in the middle. rate.Looming stimuli: no space between rate and Looming.

2) Discussion, 7th (starting "In addition to exerting"), 8th (starting "Other studies"), and 9th paragraphs (starting "In addition to modulating"). These paragraphs are all speculations. They are overly long, and thus need to be shortened.

3) Methods, 4th paragraph, in the middle esca pes

4) Methods, the paragraph of "Calcium imaging experiments." In the middle, the words of "inter-stimulus-interval" and "sequence" are separated ("sequence" starts as a new line). This need to be fixed.

*Reviewer #2 (Recommendations for the authors):*

1. Figure 1e: What is the Y-axis label of the graph? Density? Did Density = the number of individuals within the interval / the number of individuals in all intervals? If so, why is the sum of density in all intervals > 1?

2. Figure 2: First, labels c and d are not aligned with the corresponding figure. I am confused that when ISI=180s, each parameter still has a certain decline (response probability, peak tail amplitude, or response trial number), indicating there may be some kind of neurons to maintain the process of habituation, the maintenance effect is stronger when ISI=10s, but weak when ISI=180s, do you think this process is all involved by transient E/I neurons receiving visual information?

3. Figure 3: Figure c lacks statistical significance analysis. Figure d Y-axis label "relativ e" should be "relative". Figure d DM group: do fish escape after 4-6 seconds of the dimming stimulation also count as responding to this stimulation? This may make your DM group escape probability higher in Figure b.

4. Figure 4: The dimming stimuli increase the response probability of the next dimming but decrease the probability of the next looming in Figure a, showing that the dimming stimulus strengthens the overall dimming feature components but weakens the overall spatial expansion feature components of a dark looming stimulus for escape. The difference between chessboard and looming is whether there is an overall luminance change, but in Figure c, compared with CB10, DL1CB only incorporated an overall dimming feature, which causes an increase in the response probability, showing the dimming stimulus strengthens the overall spatial expansion feature component. There is a certain contradiction between Figure a and Figure c unless it can be proved that the overall spatial expansion features between looming and chessboard are different, but this will have a greater impact on the overall article.

5. Figure 5: Why cluster 5 is called "looming sensitive" instead of cluster 4 in Supplementary Figure 3, cluster 4 contains both dimming and expansion features. If you want to know the impact of dimming on the looming response, why not analyze the correlation between cluster 4 and cluster 8, and then use cluster 1 as a control?

6. Figure 6: Why do "potentiating" inhibitory neurons rather than "depressing" excitatory DS neurons exert a pivotal role in the escape circuit, obviously, the proportion of the latter (33%) is much higher than that of the former (8%). Do GABAegric DSPOT and DSDEP neurons have different spatial distribution preferences?

7. Figure 7: The same question as those in Figure 6, if DS neurons are upstream of LS neurons, the increase of inhibition or the decrease of excitability can lead to similar calcium reactions in Figures c and d.

8. Figure 8-9: Lack of circuit level proof for the sufficiency and necessity of GABAergic DSPOT neurons to habituation. Spatial distribution and response properties of binocular and monocular LS and DS neurons are not the key questions of this article, the correlation between LS and DS E/I neurons is.

Other comments

1. Figures 1 and 2 can be combined into a single figure, indicating habituation can be effectively done in this experimental environment.

2. Figures 3 and 4 can be integrated into one figure, exploring the roles of the different looming components in habituation.

3. Figures 5, 6 and 8 can be combined into a single figure, whole-brain analysis looming response properties.

4. Figures 2: The y-axis can be increased so that the trend of habituation can be seen more clearly.

5. Maybe GABAergic DSPOT neurons can be labeled with dyes or viruses in the future, so the whole process of habituation from the perspective of the circuitry level can be seen more clearly.

---

## [Author Response]

Essential revisions:The strength of the present work is that the authors discovered a subpopulation of inhibitory neurons, that is potentiated during repetitive visual stimuli and control the habituation of visually evoked escape behavior. This is a great finding and should be shared with the community. I see the following weakness in the manuscript.1) The lack of any further experiments on "checkerboard stimuli".The authors emphasize the existence of another pathway for habituation induced by "checkerboard stimuli", but this has not been characterized further.

We acknowledge that the checkerboard pathway is not examined in our study and we explicitly have included this now in the discussion. We believe that several other studies address this aspect quite thoroughly. Mancienne et al. for example find, in a similar assay, little contribution of the dimming stimuli to habituation, likely because of different boundary conditions, which suggests that their study emphasizes the “checkerboard pathway” at the expense of inhibition by dimming. We believe that this specific checkerboard habituation channel is selective for just that stimulus; and the habituation process is likely explained by an activity dependent mechanism that’s intrinsic to that feedforward pathway. We now also state in the manuscript that the preferred approach to uncover such mechanisms requires molecular and cell biological tools (Randlett et al. 2019) that are not readily at our disposal and would take this manuscript into a very different direction.

2) No connectivity between DS and LS was shown.Although the authors discovered the DS neurons, which are potentiated during repeated stimuli and inhibit LS neurons, the actual connectivity has not been demonstrated.

We can see three different ways of how to cleanly approach this issue.

1) Targeted ablation of DSpot neurons predicts that habituation in LS neurons would be compromised.

2) Viral tracing could be performed to uncover specific connectivity patterns.

3) Correlated Light and Electron Microscopy (CLEM) could be performed to obtain an explicit wiring diagram and thereby confirm our hypothesis.

We note that all three options are not feasible within a reasonable timeframe, mostly because the required technology is not yet mature enough. Targeted ablations could be performed with currently available techniques, however the distributed and sparse nature of the DSpot neurons makes it unlikely that we can target a sufficiently large fraction of these neurons to get a plausible result. We, now discuss this more explicitly in the text and point out that such experiments are well suited for future studies.

We would like you to respond to the reviewer’s comments below.Reviewer #1 (Recommendations for the authors):Specific comments:1) Results, the first paragraph, in the middle. Rate.Looming stimuli: no space between rate and Looming.

Thanks for catching this typo, which we have now fixed.

2) Discussion, 7^th^ (starting “In addition to exerting”), 8^th^ (starting “Other studies”), and 9^th^ paragraphs (starting “In addition to modulating”). These paragraphs are all speculations. They are overly long, and thus need to be shortened.

We have now shortened these paragraphs as suggested.

3) Methods, 4^th^ paragraph, in the middle esca pes

Thanks and we have fixed this typo.

4) Methods, the paragraph of “Calcium imaging experiments.” In the middle, the words of “inter-stimulus-interval” and “sequence” are separated (“sequence” starts as a new line). This need to be fixed.

Indeed the word sequence is not needed in this phrase. We have now removed it from the sentence.

Reviewer #2 (Recommendations for the authors):1. Figure 1e: What is the Y-axis label of the graph? Density? Did Density = the number of individuals within the interval / the number of individuals in all intervals? If so, why is the sum of density in all intervals > 1?

Yes, the label is Density, which we have now added to the figure. The sum of density is indeed 1, as the binwidth=0.5 mm. The density panels have now been moved to Supplemental Figure 1 c,d.

2. Figure 2: First, labels c and d are not aligned with the corresponding figure. I am confused that when ISI=180s, each parameter still has a certain decline (response probability, peak tail amplitude, or response trial number), indicating there may be some kind of neurons to maintain the process of habituation, the maintenance effect is stronger when ISI=10s, but weak when ISI=180s, do you think this process is all involved by transient E/I neurons receiving visual information?

We have adjusted the location of the labels.

We agree with the referee that it is surprising that an ISI as long as 180 seconds still leads to a significant habituation effect. And we believe that the decline in response probability over such timescales is indeed the behavioral manifestation of a reduction in sensory excitation to downstream motor networks. Specifically, our model predicts that the inhibitory DSpot neurons are responsible for this maintenance; in other words it predicts that the time over which these neurons stay potentiated lasts significantly longer than 180 seconds. Indeed, we find that the DSpot neurons stay potentiated, even in the absence of stimulation, for a time window of up to ten minutes.

3. Figure 3: Figure c lacks statistical significance analysis. Figure d Y-axis label "relativ e" should be "relative". Figure d DM group: do fish escape after 4-6 seconds of the dimming stimulation also count as responding to this stimulation? This may make your DM group escape probability higher in Figure b.

(a,b) We had not added the p values before, as they were not significant. These panels are in fact removed from the current Figure 2 in the process of merging Figures 3 and 4, to save space as they did not convey data that were important to our analysis.

c) We thank the reviewer for the comment and note that the resulting reanalysis led us to introduce some significant changes. Specifically, we now subtract the spontaneous swim rate of the animals from the observed bout rate in the presence of a stimulus. This has several consequences which change some of the findings but none of the overall conclusions. Most importantly, subtracting this baseline rate from the stimulus evoked bout rates removed the behavioral effect of a dimming stimulus. In other words, the updated results are that fish show no significant response to a dimming stimulus. This new insight is actually better aligned with published results (Mancienne et al., Dunn et al., Temizer et al), and, importantly, does not take away from the finding that exposure to dimming stimuli plays a central role in habituating the animal and the looming sensitive neurons.

Furthermore, this shows that the strong modulatory effect of dimming stimuli occurs in the absence of any overt behavior. Of course, if there are no responses to dimming stimuli, the “response time relative to collision” analysis for dimming becomes obsolete.

In the effort to address the reviewers suggestion to merge figures 3 and 4, we have now removed panels c and d of figure 3 as they no longer provide information critical to our analysis.

4. Figure 4: The dimming stimuli increase the response probability of the next dimming but decrease the probability of the next looming in Figure a, showing that the dimming stimulus strengthens the overall dimming feature components but weakens the overall spatial expansion feature components of a dark looming stimulus for escape. The difference between chessboard and looming is whether there is an overall luminance change, but in Figure c, compared with CB10, DL1CB only incorporated an overall dimming feature, which causes an increase in the response probability, showing the dimming stimulus strengthens the overall spatial expansion feature component. There is a certain contradiction between Figure a and Figure c unless it can be proved that the overall spatial expansion features between looming and chessboard are different, but this will have a greater impact on the overall article.

We understand the referee’s confusion and concern and apologize for the lack of clarity. Our new and improved analysis revealed two novel aspects that hopefully resolve these issues.

1) The lack of responsiveness to dimming stimuli in fact makes the analysis shown in Figure 4 (Now Figure 2c) much more straightforward. Since the effect of the exposure to dimming stimuli is latent (current Figure 2b); we now simply plot the effects of pre-exposure to dimming and confirm our finding that dimming stimuli, indeed, remove the ability of the fish to respond to a looming stimulus, even in the absence of an overt behavioral response to dimming.

2) We agree with the referee’s astute observation that the recovery of the behavior in response to a dark loom (after being habituated by checkerboard) requires an explanation; especially if we cannot attribute this recovered response to the residual dimming aspect contained in the dark-loom but not the checkerboard. A more parsimonious explanation is that the dark-loom stimulus contains homogeneous expansion features (Gabbiani et al. 2001) that are absent in the “patchy” or heterogenous checkerboard; and that it is these homogenous features which are responsible for driving the recovered response because they are protected from habituation during repeated, heterogenous, checkerboard exposure.

In short, we propose that the lack of an effect of “pre-exposure” to checkerboard on the looming response indicates and confirms our hypothesis that the “checkerboard” channel habituates independently of the dark looming channel.

All of these aspects are now discussed in the revised manuscript.

5. Figure 5: Why cluster 5 is called "looming sensitive" instead of cluster 4 in Supplementary Figure 3, cluster 4 contains both dimming and expansion features. If you want to know the impact of dimming on the looming response, why not analyze the correlation between cluster 4 and cluster 8, and then use cluster 1 as a control?

Cluster 5 is the universal “looming sensitive” channel that responds to both, checkerboard and dark loom, regardless of the dimming aspect of the stimulus. Channel 4 only responds to dark loom and ignores checkerboard.

If you want to know the impact of dimming on the looming response, why not analyze the correlation between cluster 4 and cluster 8, and then use cluster 1 as a control?

We thank the referee for the insightful comment and we have considered following up on the suggested analyses. However, the assumption that channel 8 isolates the dimming component and that channel 1 isolates the expansion component is not justified. The issue with channel 8 and channel 1 is that we know that they don’t actually isolate the dimming or expansion component respectively, because they ignore these aspects when presented in a dark looming stimulus. As such they probably play a specialized role in processing expansion or dimming signals that occur independently of dark looming. Either way, these are highly specialized channels that must contain non-linear processing features and are therefore not well suited for basic control experiments that test the impact of dimming on the looming response.

Therefore, we chose cluster 5, which together with cluster 12 allows teasing apart the interaction between dimming and looming aspects of the stimulus because they are both active in response to DL stimuli. Cluster 8, on the other hand, is not responsive to DL stimuli and therefore, could not have an effect on the looming response and does not allow doing correlation analysis with either cluster 4 or 5.

6. Figure 6: Why do "potentiating" inhibitory neurons rather than "depressing" excitatory DS neurons exert a pivotal role in the escape circuit, obviously, the proportion of the latter (33%) is much higher than that of the former (8%). Do GABAegric DSPOT and DSDEP neurons have different spatial distribution preferences?

This is a valid and useful suggestion and we have, indeed, considered such an alternate model that includes depressing excitatory DS neurons as the pivotal element. However, such an implementation was less compatible with experimental data because such a model requires an excitatory drive between dimming selective units and cluster 5, and inspecting the response properties of cluster 5 revealed that such connections do not exist: a dimming stimulus does not evoke any responses in those neurons.

7. Figure 7: The same question as those in Figure 6, if DS neurons are upstream of LS neurons, the increase of inhibition or the decrease of excitability can lead to similar calcium reactions in Figures c and d.

Please see our response to point #6 above.

8. Figure 8-9: Lack of circuit level proof for the sufficiency and necessity of GABAergic DSPOT neurons to habituation. Spatial distribution and response properties of binocular and monocular LS and DS neurons are not the key questions of this article, the correlation between LS and DS E/I neurons is.

The referee is correct, we have no proof of necessity nor sufficiency for the existence of a connection between inhibitory DSPOT and LS neurons. This would require causal neuroscience experiments including ablations, connectomics or optogenetics, all of which are hard to do because of the distributed nature and relative sparsity of the critical neuronal populations. It is unlikely that we could target a sufficiently large fraction with existing technology to expect a significant effect.

We now address these shortcomings explicitly in the discussion.

Other comments1. Figures 1 and 2 can be combined into a single figure, indicating habituation can be effectively done in this experimental environment.

Thanks for this helpful suggestion. We have now merged Figures 1 and 2 into the current Figure 1.

2. Figures 3 and 4 can be integrated into one figure, exploring the roles of the different looming components in habituation.

We have now merged the two figures (now Figure 2) and removed panels that did not contain information critical to our analysis (see above).

3. Figures 5, 6 and 8 can be combined into a single figure, whole-brain analysis looming response properties.

We have combined figures 5 and 6 (now Figure 3) but believe that combining Figure 8 with those will result in issues in the flow of the manuscript (by passing Figure 7). So Figures 8-9 remain as before and now re-numbered as Figures 4-6.

4. Figures 2: The y-axis can be increased so that the trend of habituation can be seen more clearly.

Thanks for this suggestion. We have now increased the y-axis so the habituation trend can be seen more clearly (now in Figure 1).

5. Maybe GABAergic DSPOT neurons can be labeled with dyes or viruses in the future, so the whole process of habituation from the perspective of the circuitry level can be seen more clearly.

That would, indeed, be a wonderful addition to our experiments, and it opens the door for a whole series of interesting future studies.